# Kernel Observers: Systems-Theoretic Modeling and Inference of Spatiotemporally Evolving Processes

**Hassan A. Kingravi**
Pindrop
Atlanta, GA 30308
hkingravi@pindrop.com

**Harshal Maske and Girish Chowdhary**
University of Illinois at Urbana Champaign
Urbana, IL 61801
hmaske2@illinois.edu, girishc@illinois.edu

## Abstract

We consider the problem of estimating the latent state of a spatiotemporally evolving continuous function using very few sensor measurements. We show that layering a dynamical systems prior over temporal evolution of weights of a kernel model is a valid approach to spatiotemporal modeling, and that it does not require the design of complex nonstationary kernels. Furthermore, we show that such a differentially constrained predictive model can be utilized to determine sensing locations that guarantee that the hidden state of the phenomena can be recovered with very few measurements. We provide sufficient conditions on the number and spatial location of samples required to guarantee state recovery, and provide a lower bound on the minimum number of samples required to robustly infer the hidden states. Our approach outperforms existing methods in numerical experiments.

## 1 Introduction

Modeling of large-scale stochastic phenomena with both spatial and temporal (spatiotemporal) evolution is a fundamental problem in the applied sciences and social networks. The spatial and temporal evolution in such domains is constrained by stochastic partial differential equations, whose structure and parameters may be time-varying and unknown. While modeling spatiotemporal phenomena has traditionally been the province of the field of geostatistics, it has in recent years gained more attention in the machine learning community [2]. The data-driven models developed through machine learning techniques provide a way to capture complex spatiotemporal phenomena that are not easily modeled by first-principles alone, such as stochastic partial differential equations.

In the machine learning community, kernel methods represent a class of extremely well-studied and powerful methods for inference in spatial domains; in these techniques, correlations between the input variables are encoded through a covariance kernel, and the model is formed through a linear weighted combination of the kernels [14]. In recent years, kernel methods have been applied to spatiotemporal modeling with varying degrees of success [2, 14]. Many recent techniques in spatiotemporal modeling have focused on nonstationary covariance kernel design and associated hyperparameter learning algorithms [4, 7, 12]. The main benefit of careful design of covariance kernels over approaches that simply include time as an additional input variable is that they can account for intricate spatiotemporal couplings. However, there are two key challenges with these approaches: the first is ensuring the scalability of the model to large scale phenomena, which manifests due to the fact that the hyperparameter optimization problem is not convex in general, leading to methods that are difficult to implement, susceptible to local minima, and that can become computationally intractable for large datasets. In addition to the challenge of modeling spatiotemporally varying processes, we are interested in addressing the second very important, and widely unaddressed challenge: given a predictive model of the spatiotemporal phenomena, how can the current latent state of the phenomena be estimated using as few sensor measurements as possible? This is called the *monitoring problem*.

Monitoring a spatiotemporal phenomenon is concerned with estimating its current state, predicting its future evolution, and inferring the initial conditions utilizing limited sensor measurements. The key challenges here manifest due to the fact that it is typically infeasible or expensive to deploy sensors at a large scale across vast spatial domains. To minimize the number of sensors deployed, a predictive data-driven model of the spatiotemporal evolution could be learned from historic datasets or through remote sensing (e.g. satellite, radar) datasets. Then, to monitor the phenomenon, the key problem would boil down to reliably and quickly estimating the evolving latent state of the phenomena utilizing measurements from very few sampling locations.

In this paper, we present an alternative perspective on solving the spatiotemporal monitoring problem that brings together kernel-based modeling, systems theory, and Bayesian filtering. Our main contributions are two-fold: first, we demonstrate that spatiotemporal functional evolution can be modeled using stationary kernels with a linear dynamical systems layer on their mixing weights. In other words, the model proposed here posits *differential constraints*, embodied as a linear dynamical system, on the spatiotemporal evolution of a kernel based models, such as Gaussian Processes. This approach does not necessarily require the design of complex spatiotemporal kernels, and can accommodate positive-definite kernels on any domain on which it's possible to define them, which includes non-Euclidean domains such as Riemannian manifolds, strings, graphs and images [6]. Second, we show that the model can be utilized to determine sensing locations that guarantee that the hidden states of functional evolution can be estimated using a Bayesian state-estimator with very few measurements. We provide sufficient conditions on the number and location of sensor measurements required and prove non-conservative lower bounds on the minimum number of sampling locations. The validity of the presented model and sensing techniques is corroborated using synthetic and large real datasets.

## 1.1 Related Work

There is a large body of literature on spatiotemporal modeling in geostatistics where specific process-dependent kernels can be used [17, 2]. From the machine learning perspective, a naive approach is to utilize both spatial and temporal variables as inputs to a Mercer kernel [10]. However, this technique leads to an ever-growing kernel dictionary. Furthermore, constraining the dictionary size or utilizing a moving window will occlude learning of long-term patterns. Periodic or nonstationary covariance functions and nonlinear transformations have been proposed to address this issue [7, 14]. Work focusing on nonseparable and nonstationary covariance kernels seeks to design kernels optimized for environment-specific dynamics, and to tune their hyperparameters in local regions of the input space. Seminal work in [5] proposes a process convolution approach for space-time modeling. This model captures nonstationary structure by allowing the convolution kernel to vary across the input space. This approach can be extended to a class of nonstationary covariance functions, thereby allowing the use of a Gaussian process (GP) framework, as shown in [9]. However, since this model's hyperparameters are inferred using MCMC integration, its application has been limited to smaller datasets. To overcome this limitation, [12] proposes to use the mean estimates of a second isotropic GP (defined over latent length scales) to parameterize the nonstationary covariances. Finally, [4] considers nonistropic variation across different dimension of input space for the second GP as opposed to isotropic variation by [12]. Issues with this line of approach include the nonconvexity of the hyperparameter optimization problem and the fact that selection of an appropriate nonstationary covariance function for the task at hand is a nontrivial design decision (as noted in [16]).

Apart from directly modeling the covariance function using additional latent GPs, there exist several other approaches for specifying nonstationary GP models. One approach maps the nonstationary spatial process into a latent space, in which the problem becomes approximately stationary [15]. Along similar lines, [11] extends the input space by adding latent variables, which allows the model to capture nonstationarity in original space. Both these approaches require MCMC sampling for inference, and as such are subject to the limitations mentioned in the preceding paragraph.

A geostatistics approach that finds dynamical transition models on the linear combination of weights of a parameterized model [2, 8] is advantageous when the spatial and temporal dynamics are hierarchically separated, leading to a convex learning problem. As a result, complex nonstationary kernels are often not necessary (although they can be accommodated). The approach presented in this paper aligns closely with this vein of work. A system theoretic study of this viewpoint enables the fundamental contributions of the paper, which are 1) allowing for inference on more general domains with a larger class of basis functions than those typically considered in the geostatistics community,

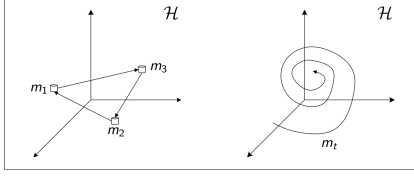

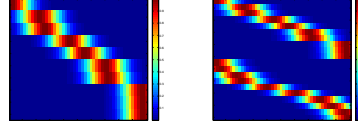

(a) 1-shaded (Def. 1)(b) 2-shaded (Eq. (4))

Figure 1: Two types of Hilbert space evolutions. Left: discrete switches in RKHS $\mathcal{H}$; Right: smooth evolution in $\mathcal{H}$.

Figure 2: Shaded observation matrices for dictionary of atoms.

and 2) quantifying the minimum number of measurements required to estimate the state of functional evolution.

It should be noted that the contribution of the paper concerning sensor placement is to provide sufficient conditions for monitoring rather than optimization of the placement locations, hence a comparison with these approaches is not considered in the experiments.

## 2 Kernel Observers

This section outlines our modeling framework and presents theoretical results associated with the number of sampling locations required for monitoring functional evolution.

### 2.1 Problem Formulation

We focus on predictive inference of a time-varying stochastic process, whose mean $f$ evolves temporally as $f_{\tau+1} \sim \mathbb{F}(f_\tau, \eta_\tau)$, where $\mathbb{F}$ is a distribution varying with time $\tau$ and exogenous inputs $\eta$. Our approach builds on the fact that in several cases, temporal evolution can be hierarchically separated from spatial functional evolution. A classical and quite general example of this is the *abstract evolution equation* (AEO), which can be defined as the evolution of a function $u$ embedded in a Banach space $\mathcal{B}$: $\dot{u}(t) = \mathcal{L}u(t)$, subject to $u(0) = u_0$, and $\mathcal{L} : \mathcal{B} \to \mathcal{B}$ determines spatiotemporal transitions of $u \in \mathcal{B}$ [1]. This model of spatiotemporal evolution is very general (AEOs, for example, model many PDEs), but working in Banach spaces can be computationally taxing. A simple way to make the approach computationally realizable is to place restrictions on $\mathcal{B}$: in particular, we restrict the sequence $f_\tau$ to lie in a reproducing kernel Hilbert space (RKHS), the theory of which provides powerful tools for generating flexible classes of functions with relative ease [14]. In a kernel-based model, $k : \Omega \times \Omega \to \mathbb{R}$ is a positive-definite Mercer kernel on a domain $\Omega$ that models the covariance between any two points in the input space, and implies the existence of a smooth map $\psi : \Omega \to \mathcal{H}$, where $\mathcal{H}$ is an RKHS with the property $k(x,y) = \langle \psi(x), \psi(y) \rangle_\mathcal{H}$. The key insight behind the proposed model is that spatiotemporal evolution in the input domain corresponds to temporal evolution of the mixing weights of a kernel model alone in the functional domain. Therefore, $f_\tau$ can be modeled by tracing the evolution of its mean embedded in a RKHS using switched ordinary differential equations (ODE) when the evolution is continuous, or switched difference equations when it is discrete (Figure 1). The advantage of this approach is that it allows us to utilize powerful ideas from systems theory for deriving necessary and sufficient conditions for spatiotemporal monitoring. In this paper, we restrict our attention to the class of functional evolutions $\mathbb{F}$ defined by linear Markovian transitions in an RKHS. While extension to the nonlinear case is possible (and non-trivial), it is not pursued in this paper to help ease the exposition of the key ideas. The class of linear transitions in RKHS is rich enough to model many real-world datasets, as suggested by our experiments.

Let $y_\tau \in \mathbb{R}^N$ be the measurements of the function available from $N$ sensors at time $\tau$, $\mathcal{A} : \mathcal{H} \to \mathcal{H}$ be a linear transition operator in the RKHS $\mathcal{H}$, and $\mathcal{K} : \mathcal{H} \to \mathbb{R}^N$ be a linear measurement operator. The model for the functional evolution and measurement studied in this paper is:

$$f_{\tau+1} = \mathcal{A}f_\tau + \eta_\tau, \quad y_\tau = \mathcal{K}f_\tau + \zeta_\tau, \tag{1}$$

where $\eta_\tau$ is a zero-mean stochastic process in $\mathcal{H}$, and $\zeta_\tau$ is a Wiener process in $\mathbb{R}^N$. Classical treatments of kernel methods emphasize that for most kernels, the feature map $\psi$ is unknown, and possibly infinite-dimensional; this forces practioners to work in the dual space of $\mathcal{H}$, whose dimensionality is the number of samples in the dataset being modeled. This conventional wisdom precludes the use of kernel methods for most tasks involving modern datasets, which may have

millions and sometimes billions of samples [13]. An alternative is to work with a feature map $\widehat{\psi}(x) := \left[\,\widehat{\psi}_1(x) \, \cdots \, \widehat{\psi}_M(x)\,\right]$ to an approximate feature space $\widehat{\mathcal{H}}$, with the property that for every element $f \in \mathcal{H}$, $\exists \widehat{f} \in \widehat{\mathcal{H}}$ and an $\epsilon > 0$ s.t. $\|f - \widehat{f}\| < \epsilon$ for an appropriate function norm. A few such approximations are listed below.

**Dictionary of atoms**　Let $\Omega$ be compact. Given points $\mathcal{C} = \{c_1, \ldots, c_M\}$, $c_i \in \Omega$, we have a dictionary of atoms $\mathcal{F}^{\mathcal{C}} = \{\psi(c_1), \cdots, \psi(c_M)\}$, $\psi(c_i) \in \mathcal{H}$, the span of which is a strict subspace $\widehat{\mathcal{H}}$ of the RKHS $\mathcal{H}$ generated by the kernel. Here,

$$\widehat{\psi}_i(x) := \langle \psi(x), \psi(c_i) \rangle_{\mathcal{H}} = k(x, c_i) \tag{2}$$

**Low-rank approximations**　Let $\Omega$ be compact, let $\mathcal{C} = \{c_1, \ldots, c_M\}$, $c_i \in \Omega$, and let $K \in \mathbb{R}^{M \times M}$, $K_{ij} := k(c_i, c_j)$ be the Gram matrix computed from $\mathcal{C}$. This matrix can be diagonalized to compute approximations $(\widehat{\lambda}_i, \widehat{\phi}_i(x))$ of the eigenvalues and eigenfunctions $(\lambda_i, \phi_i(x))$ of the kernel [18]. These spectral quantities can then be used to compute $\widehat{\psi}_i(x) := \sqrt{\widehat{\lambda}_i}\widehat{\phi}_i(x)$.

**Random Fourier features**　Let $\Omega \subset \mathbb{R}^n$ be compact, and let $k(x, y) = e^{-\|x-y\|^2/2\sigma^2}$ be the Gaussian RBF kernel. Then random Fourier features approximate the kernel feature map as $\widehat{\psi}_\omega : \Omega \to \widehat{\mathcal{H}}$, where $\omega$ is a sample from the Fourier transform of $k(x, y)$, with the property that $k(x, y) = \mathbb{E}_\omega[\langle \widehat{\psi}_\omega(x), \widehat{\psi}_\omega(y) \rangle_{\widehat{\mathcal{H}}}]$ [13]. In this case, if $V \in \mathbb{R}^{M/2 \times n}$ is a random matrix representing the sample $\omega$, then $\widehat{\psi}_i(x) := \left[\,\frac{1}{\sqrt{M}}\sin([Vx]_i), \frac{1}{\sqrt{M}}\cos([Vx]_i)\,\right]$. Similar approximations exist for other radially symmetric and dot product kernels.

In the approximate space case, we replace the transition operator $\mathcal{A} : \mathcal{H} \to \mathcal{H}$ in (1) by $\widehat{\mathcal{A}} : \widehat{\mathcal{H}} \to \widehat{\mathcal{H}}$. This approximate regime, which trades off the flexibility of a truly nonparametric approach for computational realizability, still allows for the representation of rich phenomena, as will be seen in the sequel. The finite-dimensional evolution equations approximating (1) in dual form are

$$w_{\tau+1} = \widehat{A}w_\tau + \eta_\tau, \quad y_\tau = Kw_\tau + \zeta_\tau, \tag{3}$$

where we have matrices $\widehat{A} \in \mathbb{R}^{M \times M}$, $K \in \mathbb{R}^{N \times M}$, the vectors $w_\tau \in \mathbb{R}^M$, and where we have slightly abused notation to let $\eta_\tau$ and $\zeta_\tau$ denote their $\widehat{\mathcal{H}}$ counterparts. Here $K$ is the matrix whose rows are of the form $K_{(i)} = \widehat{\Psi}(x_i) = \left[\,\widehat{\psi}_1(x_i) \, \widehat{\psi}_2(x_i) \, \cdots \, \widehat{\psi}_M(x_i)\,\right]$. In systems-theoretic language, each row of $K$ corresponds to a *measurement* at a particular location, and the matrix itself acts as a measurement operator. We define the *generalized observability matrix* [20] as $\mathcal{O}_\Upsilon = \begin{bmatrix} K\widehat{A}^{\tau_1} \\ \cdots \\ K\widehat{A}^{\tau_L} \end{bmatrix}$ where $\Upsilon = \{\tau_1, \ldots, \tau_L\}$ are the set of instances $\tau_i$ when we apply the operator $K$. A linear system is said to be *observable* if $\mathcal{O}_\Upsilon$ has full column rank (i.e. $\mathrm{Rank}\mathcal{O}_\Upsilon = M$) for $\Upsilon = \{0, 1, \ldots, M-1\}$ [20]. Observability guarantees two critical facts: firstly, it guarantees that the state $w_0$ can be recovered exactly from a finite series of measurements $\{y_{\tau_1}, y_{\tau_2}, \ldots, y_{\tau_L}\}$; in particular, defining $y_\Upsilon = \left[y_{\tau_1}^T, y_{\tau_2}^T, \cdots, y_{\tau_L}^T\right]^T$, we have that $y_\Upsilon = \mathcal{O}_\Upsilon w_0$. Secondly, it guarantees that a feedback based *observer* can be designed such that the estimate of $w_\tau$, denoted by $\widehat{w}_\tau$, converges exponentially fast to $w_\tau$ in the limit of samples. Note that all our theoretical results assume $\widehat{A}$ is available: while we perform system identification in the experiments (Section 3.3), it is not the focus of the paper.

We are now in a position to formally state the spatiotemporal modeling and inference problem considered: given a spatiotemporally evolving system modeled using (3), choose a set of $N$ sensing locations such that even with $N \ll M$, the functional evolution of the spatiotemporal model can be estimated (which corresponds to *monitoring*) and can be predicted robustly (which corresponds to *Bayesian filtering*). Our approach to solve this problem relies on the design of the measurement operator $K$ so that the pair $(K, \widehat{A})$ is observable: any Bayesian state estimator (e.g. a Kalman filter) utilizing this pair is denoted as a **kernel observer** [1]. We will leverage the spectral decomposition of $\widehat{A}$ for this task (see §**??** in supplementary for details on spectral decomposition).

## 2.2　Main Results

In this section, we prove results concerning the observability of spatiotemporally varying functions modeled by the functional evolution and measurement equations (3) formulated in Section 2.1. In

particular, observability of the system states implies that we can recover the current state of the spatiotemporally varying function using a small number of sampling locations $N$, which allows us to 1) track the function, and 2) predict its evolution forward in time. We work with the approximation $\widehat{\mathcal{H}} \approx \mathcal{H}$: given $M$ basis functions, this implies that the dual space of $\widehat{\mathcal{H}}$ is $\mathbb{R}^M$. Proposition 1 shows that if $\widehat{A}$ has a full-rank Jordan decomposition, the observation matrix $K$ meeting a condition called *shadedness* (Definition 1) is sufficient for the system to be observable. Proposition 2 provides a lower bound on the number of sampling locations required for observability which holds for any $\widehat{A}$. Proposition 3 constructively shows the existence of an abstract measurement map $\widetilde{K}$ achieving this lower bound. Finally, since the measurement map does not have the structure of a kernel matrix, a slightly weaker sufficient condition for the observability of any $\widehat{A}$ is in Theorem 1. Proofs of all claims are in the supplementary material.

**Definition 1.** *(Shaded Observation Matrix) Given $k : \Omega \times \Omega \to \mathbb{R}$ positive-definite on a domain $\Omega$, let $\{\widehat{\psi}_1(x), \ldots, \widehat{\psi}_M(x)\}$ be the set of bases generating an approximate feature map $\widehat{\psi} : \Omega \to \widehat{\mathcal{H}}$, and let $\mathcal{X} = \{x_1, \ldots, x_N\}$, $x_i \in \Omega$. Let $K \in \mathbb{R}^{N \times M}$ be the observation matrix, where $K_{ij} := \widehat{\psi}_j(x_i)$. For each row $K_{(i)} := [\widehat{\psi}_1(x_i) \cdots \widehat{\psi}_M(x_i)]$, define the set $\mathcal{I}_{(i)} := \{\iota_1^{(i)}, \iota_2^{(i)}, \ldots, \iota_{M_i}^{(i)}\}$ to be the indices in the observation matrix row $i$ which are nonzero. Then if $\bigcup_{i \in \{1, \ldots, N\}} \mathcal{I}^{(i)} = \{1, 2, \ldots, M\}$, we denote $K$ as a* shaded observation matrix *(see Figure 2a).*

This definition seems quite abstract, so the following remark considers a more concrete example.

**Remark 1.** *Let $\widehat{\psi}$ be generated by the dictionary given by $\mathcal{C} = \{c_1, \ldots, c_M\}$, $c_i \in \Omega$. Note that since $\widehat{\psi}_j(x_i) = \langle \psi(x_i), \psi(c_j) \rangle_{\mathcal{H}} = k(x_i, c_j)$, $K$ is the kernel matrix between $\mathcal{X}$ and $\mathcal{C}$. For the kernel matrix to be shaded thus implies that there does not exist an atom $\psi(c_j)$ such that the projections $\langle \psi(x_i), \psi(c_j) \rangle_{\mathcal{H}}$ vanish for all $x_i$, $1 \leq i \leq N$. Intuitively, the shadedness property requires that the sensor locations $x_i$ are privy to information propagating from every $c_j$. As an example, note that, in principle, for the Gaussian kernel, a single row generates a shaded kernel matrix[2].*

**Proposition 1.** *Given $k : \Omega \times \Omega \to \mathbb{R}$ positive-definite on a domain $\Omega$, let $\{\widehat{\psi}_1(x), \ldots, \widehat{\psi}_M(x)\}$ be the set of bases generating an approximate feature map $\widehat{\psi} : \Omega \to \widehat{\mathcal{H}}$, and let $\mathcal{X} = \{x_1, \ldots, x_N\}$, $x_i \in \Omega$. Consider the discrete linear system on $\widehat{\mathcal{H}}$ given by the evolution and measurement equations (3). Suppose that a full-rank Jordan decomposition of $\widehat{A} \in \mathbb{R}^{M \times M}$ of the form $\widehat{A} = P \Lambda P^{-1}$ exists, where $\Lambda = [\Lambda_1 \cdots \Lambda_O]$, and there are no repeated eigenvalues. Then, given a set of time instances $\Upsilon = \{\tau_1, \tau_2, \ldots, \tau_L\}$, and a set of sampling locations $\mathcal{X} = \{x_1, \ldots, x_N\}$, the system (3) is observable if the observation matrix $K_{ij}$ is shaded according to Definition 1, $\Upsilon$ has distinct values, and $|\Upsilon| \geq M$.*

When the eigenvalues of the system matrix are repeated, it is not enough for $K$ to be shaded. In the next proposition, we take a geometric approach and utilize the rational canonical form of $\widehat{A}$ to obtain a lower bound on the number of sampling locations required. Let $r$ be the number of unique eigenvalues of $\widehat{A}$, and let $\gamma_{\lambda_i}$ denote the geometric multiplicity of eigenvalue $\lambda_i$. Then the *cyclic index* of $\widehat{A}$ is defined as $\ell = \max_{1 \leq i \leq r} \gamma_{\lambda_i}$[19] (see supplementary section **??** for details).

**Proposition 2.** *Suppose that the conditions in Proposition 1 hold, with the relaxation that the Jordan blocks $[\Lambda_1 \cdots \Lambda_O]$ may have repeated eigenvalues (i.e. $\exists \Lambda_i$ and $\Lambda_j$ s.t. $\lambda_i = \lambda_j$). Then there exist kernels $k(x, y)$ such that the lower bound $\ell$ on the number of sampling locations $N$ is given by the cyclic index of $\widehat{A}$.*

Section **??** in supplementary gives a concrete example to build intuition regarding this lower bound. We now show how to construct a matrix $\widetilde{K}$ corresponding to the lower bound $\ell$.

**Proposition 3.** *Given the conditions stated in Proposition 2, it is possible to construct a measurement map $\widetilde{K} \in \mathbb{R}^{\ell \times M}$ for the system given by (3), such that the pair $(\widetilde{K}, \widehat{A})$ is observable.*

The construction provided in the proof of Proposition 3 is utilized in Algorithm 1, which uses the rational canonical structure of $\widehat{A}$ to generate a series of vectors $v_i \in \mathbb{R}^M$, whose iterations

**Algorithm 1** Measurement Map $\widetilde{K}$

---

**Input:** $\widehat{A} \in \mathbb{R}^{M \times M}$
Compute rational canonical form, such that $C = Q^{-1}\widehat{A}^T Q$. Set $C_0 := C$, and $M_0 := M$.
**for** $i = 1$ **to** $\ell$ **do**
    Obtain MP $\alpha_i(\lambda)$ of $C_{i-1}$. This returns associated indices $\mathcal{J}^{(i)} \subset \{1, 2, \ldots, M_{i-1}\}$.
    Construct vector $v_i \in \mathbb{R}^M$ such that $\xi_{v_i}(\lambda) = \alpha_i(\lambda)$ .
    Use indices $\{1, 2, \ldots, M_{i-1}\} \setminus \mathcal{J}^{(i)}$ to select matrix $C_i$. Set $M_i := |\{1, 2, \ldots, M_{i-1}\} \setminus \mathcal{J}^{(i)}|$
**end for**
Compute $\mathring{K} = [v_1^T, v_2^T, ..., v_\ell^T]^T$
**Output:** $\widetilde{K} = \mathring{K} Q^{-1}$

---

$\{v_1, \ldots, \widehat{A}^{m_1-1}v_1, \ldots, v_\ell, \ldots, \widehat{A}^{m_\ell-1}v_\ell\}$ generate a basis for $\mathbb{R}^M$. Unfortunately, the measurement map $\widetilde{K}$, being an abstract construction unrelated to the kernel, does not directly select $\mathcal{X}$. We will show how to use the measurement map to guide a search for $\mathcal{X}$ in Remark **??**. For now, we state a sufficient condition for observability of a general system.

**Theorem 1.** *Suppose that the conditions in Proposition 1 hold, with the relaxation that the Jordan blocks $[\Lambda_1 \quad \cdots \quad \Lambda_O]$ may have repeated eigenvalues. Let $\ell$ be the cyclic index of $\widehat{A}$. Define*

$$\mathbf{K} = \left[ {}_{K^{(1)T} \ldots K^{(\ell)T}} \right]^T \tag{4}$$

*as the $\ell$-shaded matrix which consists of $\ell$ shaded matrices with the property that any subset of $\ell$ columns in the matrix are linearly independent from each other. Then system (3) is observable if $\Upsilon$ has distinct values, and $|\Upsilon| \geq M$.*

While Theorem 1 is a quite general result, the condition that any $\ell$ columns of $\mathbf{K}$ be linearly independent is a very stringent condition. One scenario where this condition can be met with minimal measurements is in the case when the feature map $\widehat{\psi}(x)$ is generated by a dictionary of atoms with the Gaussian RBF kernel evaluated at sampling locations $\{x_1, \ldots, x_N\}$ according to (2), where $x_i \in \Omega \subset \mathbb{R}^d$, and $x_i$ are sampled from a non-degenerate probability distribution on $\Omega$ such as the uniform distribution. For a semi-deterministic approach, when the dynamics matrix $\widehat{A}$ is block-diagonal, a simple heuristic is given in Remark **??** in the supplementary. Note that in practice the matrix $\widehat{A}$ needs to be inferred from measurements of the process $f_\tau$. If no assumptions are placed on $\widehat{A}$, at least $M$ sensors are required for the system identification phase. Future work will study the precise conditions under which system identification is possible with less than $M$ sensors. Finally, computing the Jordan and rational canonical forms can be computationally expensive: see the supplementary for more details. We note that the crucial step in our approach is computing the cyclic index, which gives us the minimum number of sensors that need to be deployed, the computational complexity of which is $\mathcal{O}(M^3)$. Computation of the canonical forms is required in the case we need to strictly realize the lower bound on the number of sensors.

## 3 Experimental Results

### 3.1 Sampling Locations for Synthetic Data Sets

The goal of this experiment is to investigate the dependency of the observability of system (3) on the shaded observation matrix and the lower bound presented in Proposition 2. The domain is fixed on the interval $\Omega = [0, 2\pi]$. First, we pick sets of points $\mathcal{C}^{(\iota)} = \{c_1, \ldots, c_{M_\iota}\}$, $c_j \in \Omega$, $M = 50$, and construct a dynamics matrix $A = \Lambda \in \mathbb{R}^{M \times M}$, with cyclic index 5. We pick the RBF kernel $k(x, y) = e^{-\|x-y\|^2/2\sigma^2}$, $\sigma = 0.02$. Generating samples $\mathcal{X} = \{x_1, \ldots, x_N\}$, $x_i \in \Omega$ randomly, we compute the $\ell$-shaded property and observability for this system. Figure 3a shows how shadedness is a necessary condition for observability, validating Proposition 1: the slight gap between shadedness and observability here can be explained due to numerical issues in computing the rank of $\mathcal{O}_\Upsilon$. Next, we again pick $M = 50$, but for a system with a cyclic index $\ell = 18$. We constructed the measurement map $\widetilde{K}$ using Algorithm 1, and the heuristic in Remark **??** (Algorithm 2 in the supplementary) as well as random sampling to generate the sampling locations $\mathcal{X}$. These results are presented in Figure 3b. The plot for random sampling has been averaged over 100 runs. It is evident from the plot that

observability cannot be achieved for a number of samples $N < \ell$. Clearly, the heuristic presented outperforms random sampling; note however, that our intent is not to compare the heuristic against random sampling, but to show that the lower bound $\ell$ provides decisive guidelines for selecting the number of samples while using the computationally efficient random approach.

## 3.2 Comparison With Nonstationary Kernel Methods on Real-World Data

We use two real-world datasets to evaluate and compare the kernel observer with the two different lines of approach for non-stationary kernels discussed in Section 1.1. For the Process Convolution with Local Smoothing Kernel (PCLSK) and Latent Extension of Input Space (LEIS) approaches, we compare with NOSTILL-GP [4] and [11] respectively, on the Intel Berkeley and Irish Wind datasets.

Model inference for the kernel observer involved three steps: 1) picking the Gaussian RBF kernel $k(x, y) = e^{-\|x-y\|^2/2\sigma^2}$, a search for the ideal $\sigma$ is performed for a sparse Gaussian Process model (with a fixed basis vector set $\mathcal{C}$ selected using the method in [3]. For the data set discussed in this section, the number of basis vectors were equal to the number of sensing locations in the training set, with the domain for input set defined over $\mathbb{R}^2$; 2) having obtained $\sigma$, Gaussian process inference is used to generate weight vectors for each time-step in the training set, resulting in the sequence $w_\tau, \tau \in \{1, \ldots, T\}$; 3) matrix least-squares is applied to this sequence to infer $\widehat{A}$ (Algorithm 3 in the supplementary). For prediction in the autonomous setup, $\widehat{A}$ is used to propagate the state $w_\tau$ forward to make predictions with no feedback, and in the observer setup, a Kalman filter (Algorithm 4 in the supplementary) with $N$ determined using Proposition 2, and locations picked randomly, is used to propagate $w_\tau$ forward to make predictions. We also compare with a baseline GP (denoted by 'original GP'), which is the sparse GP model trained using all of the available data.

Our first dataset, the Intel Berkeley research lab temperature data, consists of 50 wireless temperature sensors in indoor laboratory region spanning 40.5 meters in length and 31 meters in width[3]. Training data consists of temperature data on March 6th 2004 at intervals of 20 minutes (beginning 00:20 hrs) which totals to 72 timesteps. Testing is performed over another 72 timesteps beginning 12:20 hrs of the same day. Out of 50 locations, we uniformly selected 25 locations each for training and testing purposes. Results of the prediction error are shown in box-plot form in Figure 4a and as a time-series in Figure 4b, note that 'Auto' refers to autonomous set up. Here, the cyclic index of $\widehat{A}$ was determined to be 2, so $N$ was set to 2 for the kernel observer with feedback. Note that here, even the autonomous kernel observer outperforms PCLSK and LEIS overall, and the kernel observer with feedback $N = 2$ does so significantly, which is why we did not include results with N > 2.

The second dataset is the Irish wind dataset, consisting of daily average wind speed data collected from year 1961 to 1978 at 12 meteorological stations in the Republic of Ireland[4]. The prediction error is in box-plot form in Figure 5a and as a time-series in Figure 5b. Again, the cyclic index of $\widehat{A}$ was determined to be 2. In this case, the autonomous kernel observer's performance is comparable to PCLSK and LEIS, while the kernel observer with feedback with $N = 2$ again outperforms all other methods. Table **??** in the supplementary reports the total training and prediction times associated with PCLSK, LEIS, and the kernel observer. We observed that 1) the kernel observer is an order of magnitude faster, and 2) even for small sets, competing methods did not scale well.

## 3.3 Prediction of Global Ocean Surface Temperature

We analyzed the feasibility of our approach on a large dataset from the National Oceanographic Data Center: the $4$ km AVHRR Pathfinder project, which is a satellite monitoring global ocean surface temperature (Fig. 6a). This dataset is challenging, with measurements at over 37 million possible coordinates, but with only around 3-4 million measurements available per day, leading to a lot of missing data. The goal was to learn the day and night temperature models on data from the year 2011, and to monitor thereafter for 2012. Success in monitoring would demonstrate two things: 1) the modeling process can capture spatiotemporal trends that generalize across years, and 2) the observer framework allows us to infer the state using a number of measurements that are an order of magnitude fewer than available. Note that due to the size of the dataset and the high computational requirements of the nonstationary kernel methods, a comparison with them was not pursued. To build the autonomous kernel observer and general kernel observer models, we followed the same procedure outlined in Section 3.2, but with $\mathcal{C} = \{c_1, \ldots, c_M\}$, $c_j \in \mathbb{R}^2$, $|\mathcal{C}| = 300$. Cyclic

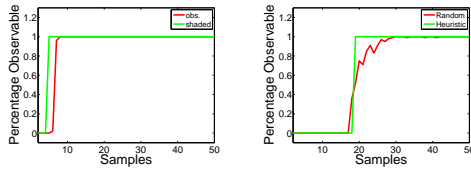

(a) Shaded vs. observability (b) Heuristic vs. random

Figure 3: Kernel observability results.

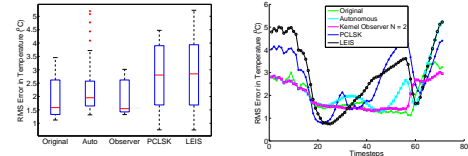

(a) Error (boxplot)  (b) Error (time-series)

Figure 4: Comparison of kernel observer to PCLSK and LEIS methods on Intel dataset.

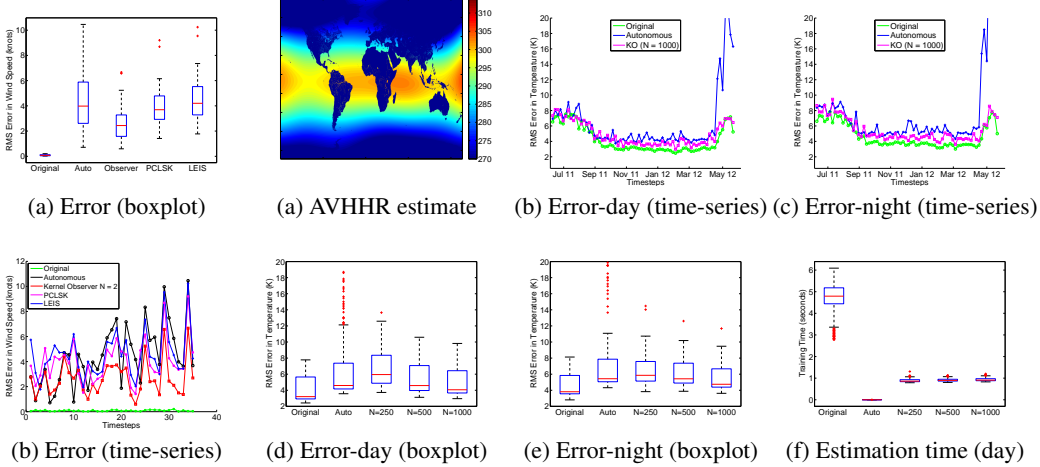

(a) Error (boxplot)  (a) AVHHR estimate  (b) Error-day (time-series)  (c) Error-night (time-series)

(b) Error (time-series)  (d) Error-day (boxplot)  (e) Error-night (boxplot)  (f) Estimation time (day)

Figure 5: Irish Wind  Figure 6: Performance of the kernel observer over AVVHR satellite 2011-12 data with different numbers of observation locations.

index of $\widehat{A}$ was determined to be 250 and hence the Kalman filter for kernel observer model using $N \in \{250, 500, 1000\}$ at random locations was utilized to track the system state given a random initial condition $w_0$. As a fair baseline, the observers are compared to training a sparse GP model (labeled 'original') on approximately $400,000$ measurements per day. Figures 6b and 6c compare the autonomous and feedback approach with $1,000$ samples to the baseline GP; here, it can be seen that the autonomous does well in the beginning, but then incurs an unacceptable amount of error when the time series goes into 2012, i.e. where the model has not seen any training data, whereas KO does well throughout. Figures 6d and 6e show a comparison of the RMS error of estimated values from the real data. This figure shows the trend of the observer getting better state estimates as a function of the number of sensing locations $N$ [5]. Finally, the prediction time of KO is much less than retraining the model every time step, as shown in Figure 6f.

# 4 Conclusions

This paper presented a new approach to the problem of monitoring complex spatiotemporally evolving phenomena with limited sensors. Unlike most Neural Network or Kernel based models, the presented approach inherently incorporates differential constraints on the spatiotemporal evolution of the mixing weights of a kernel model. In addition to providing an elegant and efficient model, the main benefit of the inclusion of the differential constraint in the model synthesis is that it allowed the derivation of fundamental results concerning the minimum number of sampling locations required, and the identification of correlations in the spatiotemporal evolution, by building upon the rich literature in systems theory. These results are non-conservative, and as such provide direct guidance in ensuring robust real-world predictive inference with distributed sensor networks.

### Acknowledgment

This work was supported by AFOSR grant #FA9550-15-1-0146.

## Footnotes

[1]In the case where no measurements are taken, for the sake of consistency, we denote the state estimator as an **autonomous kernel observer**, despite this being something of an oxymoron.

[2]However, in this case, the matrix can have many entries that are extremely close to zero, and will probably be very ill-conditioned.

[3] http://db.csail.mit.edu/labdata/labdata.html

[4] http://lib.stat.cmu.edu/datasets/wind.desc

[5]Note that we checked the performance of training a GP with only $1,000$ samples as a control, but the average error was about 10 Kelvins, i.e. much worse than KO.

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
