[Supplementary Material · nips_supplementary.pdf]

# Kernel Observers: Supplementary Material

**Hassan A. Kingravi**
Pindrop
Atlanta, GA 30308
`hkingravi@pindrop.com`

**Harshal Maske and Girish Chowdhary**
Dept. of Agricultural and Biological Engineering
University of Illinois at Urbana Champaign
Urbana, IL 61801
`hmaske2@illinois.edu, girishc@illinois.edu`

## 1 Preliminaries on Rational Canonical Structure

We take a geometric approach towards the choice of sampling locations for inferring $w_\tau$ in (3). To do so, we utilize the Frobenius canonical form and Jordan canonical decomposition of $\mathcal{A}$ [4]. We use the notation $\mathcal{V}$, with $\dim(\mathcal{V}) = M$, to emphasize the fact that these theorems hold for any finite-dimensional vector space. The linear operator $\mathcal{A} : \mathcal{V} \to \mathcal{V}$ has a characteristic polynomial $\pi(\lambda)$ such that $\pi(\mathcal{A}) = 0$ by the Cayley-Hamilton theorem. The minimal polynomial (MP) of $\mathcal{A}$ is the monic polynomial $\alpha(\lambda)$ of least degree (denoted by $\deg(\cdot)$) such that $\alpha(\lambda) = a_0 + a_1\lambda + \cdots + \lambda^{\deg(\alpha)} = 0$, and $\alpha(\mathcal{A}) = a_0 I + a_1 \mathcal{A} + \cdots + \mathcal{A}^{\deg(\alpha)} = 0$. The MP is unique and divides $\pi(\lambda)$, so that $\deg(\alpha) \le \deg(\pi)$. The MP of a vector $v \in \mathcal{V}$ *relative to* $\mathcal{A}$ is the unique monic polynomial $\xi_v$ of least degree such that $\xi_v(\mathcal{A})v = a_0 v + a_1 \mathcal{A}v + \cdots + \mathcal{A}^{\deg(\alpha)}v = 0$. If $\deg(\alpha) = M$, then $\mathcal{A}$ is said to be *cyclic* and there exists $v \in \mathcal{V}$, such that the vectors $\{v, \mathcal{A}v, \ldots, \mathcal{A}^{M-1}v\}$ form a basis for $\mathcal{V}$; this is the same as saying that the pair $(v^T, \mathcal{A}^T)$ is observable.

A subspace $\mathcal{V}_\mathcal{S} \subset \mathcal{V}$ s.t. $\mathcal{A}\mathcal{V}_\mathcal{S} \subset \mathcal{V}_\mathcal{S}$ is $\mathcal{A}$-*cyclic* if $\mathcal{A}_{|\mathcal{V}_\mathcal{S}}$, the restriction of $\mathcal{A}$ to the subspace $\mathcal{V}_\mathcal{S}$, is cyclic. If $\alpha(\lambda)$ is the minimal polynomial of $\mathcal{A}$ and $\deg(\alpha) = m < M$, $\exists v \in \mathcal{V}$ such that $\{v, \mathcal{A}v, \ldots, \mathcal{A}^{m-1}v\}$ span an $m$-dimensional $\mathcal{A}$-cyclic subspace $\mathcal{V}_\mathcal{S}$, with $v$ being the *cyclic generator* of $\mathcal{V}_\mathcal{S}$. The subspace $\mathcal{V}_\mathcal{S}$ decomposes $\mathcal{V}$ relative to $\mathcal{A}$. By the rational (or Frobenius) canonical structure theorem, $\mathcal{A}$ can be successively decomposed into subspaces $\mathcal{V}_i \subset \mathcal{V}$, $i \in \{1, \ldots, \ell\}$, s.t. $\mathcal{V} = \mathcal{V}_1 \oplus \ldots \oplus \mathcal{V}_\ell$, $\mathcal{A}\mathcal{V}_i \subset \mathcal{V}_i$, and $\mathcal{A}_{|\mathcal{V}_i}, i \in \{1, \ldots, \ell\}$, are cyclic[1]. The integer $\ell$ is unique and is called the *cyclic index of* $\mathcal{A}$. One of our main results is to show that the cyclic index is a lower bound on the number of measurements required to reconstruct $w_\tau$ (see Proposition 3 and Algorithm 1).

Recall also that for any matrix $\mathcal{A} \in \mathbb{R}^{M \times M}$, $\exists P \in \mathbb{R}^{M \times M}$ invertible such that $\mathcal{A} = P\Lambda P^{-1}$, where $\Lambda$ is a unique block diagonal matrix with Jordan blocks with $\lambda_i$ along the diagonal. If all the eigenvalues $\lambda_i$ are nonzero and real, we say the matrix has a *full-rank Jordan decomposition*.

## 2 Discussion of Theoretical Results

The systems-theoretic approach taken in this paper reveals something rather surprising: functions with complex dynamics (with a small cyclic index) can be recovered with less sensor placements than functions with simpler dynamics. Although seemingly counterintuitive, it becomes clear that this is because complex dynamics, which are characterized by a lower geometric multiplicity of the

eigenvalues, ensure that the orbit $\Theta := \{\widehat{A}w_\tau\}_{\tau \in \Upsilon}$ traverses a greater portion of $\mathbb{R}^M \equiv \widehat{\mathcal{H}}$ and thus that fewer sensors can recover more geometric information. On the other hand, in 'simpler' functional evolution, $\Theta$ evolves along strict subspaces of $\mathbb{R}^M$, and so more independent sensors are required to infer the same amount of information. Recall that cyclic index $\ell > 1$ implies that there exist spaces $\mathcal{V}_i$ s.t. $\mathbb{R}^M = \mathcal{V}_1 \oplus \cdots \oplus \mathcal{V}_\ell$, which induces the decomposition $\widehat{\mathcal{H}} = \widehat{\mathcal{H}}_1 \oplus \cdots \oplus \widehat{\mathcal{H}}_\ell$. As the simplest nontrivial case, consider $\widehat{\psi}(x)$ defined as in (2), where $\mathcal{C} = \{c_1, c_2\}$, $c_i \in \Omega$, and pick one sensor location $x_1 \in \Omega$. Let (3) be given by $\widehat{A} = \left[\begin{smallmatrix} \lambda & 0 \\ 0 & \lambda \end{smallmatrix}\right]$, $\lambda \in \mathbb{R}$ and $|\lambda| < 1$, and let the system be deterministic (i.e. $\eta_\tau, \zeta_\tau = 0$). Here, $\ell = 2$, because there exists no $v \in \mathbb{R}^2$ s.t. $\text{span}\{v, \widehat{A}v\} = \mathbb{R}^2$. For any initial condition $w_0$ we get a discrete sequence $\{w_0, \lambda w_0, \lambda^2 w_0, \dots\}$ going to zero along the 1-dimensional subspace generated by $w_0$ (i.e. $w_0$ is an eigenvector of $\widehat{A}$). Let the set of time instances $\Upsilon$ be given by $\Upsilon = \{0, 1\}$, and consider a shaded matrix $K = \left[\begin{smallmatrix} k_{11} & k_{12} \end{smallmatrix}\right]$: then the observability matrix is given by $\mathcal{O}_\Upsilon = \left[\, K^T \; (K\widehat{A})^T \,\right]^T = \left[\begin{smallmatrix} k_{11} & k_{12} \\ \lambda k_{11} & \lambda k_{12} \end{smallmatrix}\right] = \left[\begin{smallmatrix} \mathbf{k}^T \\ \lambda \mathbf{k}^T \end{smallmatrix}\right]$, which is obviously rank-deficient, and where $\left[\begin{smallmatrix} k_{11} & k_{12} \end{smallmatrix}\right]^T := \mathbf{k}$. Intuitively, we have that $\mathcal{O}_\Upsilon w_0 = \left[\begin{smallmatrix} \langle \mathbf{k}, w_0 \rangle_{\mathbb{R}^2} \\ \lambda \langle \mathbf{k}, w_0 \rangle_{\mathbb{R}^2} \end{smallmatrix}\right]$, which implies that $\mathbf{k}$ doesn't have enough geometric information to recover the initial state. Contrast this with the case when $\widehat{A} = \left[\begin{smallmatrix} \lambda & 1 \\ 0 & \lambda \end{smallmatrix}\right]$; here, $\ell = 1$, and $\mathcal{O}_\Upsilon = \left[\begin{smallmatrix} k_{11} & k_{12} \\ \lambda k_{11} & k_{11} + \lambda k_{12} \end{smallmatrix}\right]$, which is full rank for a shaded kernel matrix $K$, and hence leads to observability. This fundamental insight is be gained from considering dynamical evolutions in the structure of the model.

Another point to note is that since the collection of bases $\{\widehat{\psi}_i(x)\}_{i=1}^M$ determines the richness of the function space $\widehat{\mathcal{H}} \approx \mathcal{H}$ we operate in, it determines the fidelity of the model approximation to the true time-varying function. As a consequence, observability of the system in $\widehat{\mathcal{H}}$ refers to the best possible approximation in $\widehat{\mathcal{H}}$. The greater the number of bases, the higher the dimensionality, which results in greater model fidelity, but which may require a much greater number of measurements for state recovery. This is where the lower bounds presented in the paper are particularly useful, because they show that for functional evolutions corresponding to certain $\widehat{A}$, *the number of sensor placements are essentially independent of the dimensionality $M$*, but depend rather on the cyclic index of $\widehat{A}$.

## 3 Training and prediction times for section 3.2 of main Article

Table 1: Total training and prediction times for Figs. 4 and 5

|  | Intel Berkeley | Irish Wind |
|---|---|---|
| *Data Size (bases-timesteps)* | 25-72 | 12-36 |
| Kernel Observer | 2.1 sec | 0.1 sec |
| PCLSK | 121.4 sec | 7.0 sec |
| LEIS | 43.8 sec | 2.8 sec |

## 4 Proofs of Main Theorems

**Definition 1.** *(Shaded Observation Matrix) Given $k : \Omega \times \Omega \to \mathbb{R}$ positive-definite on a domain $\Omega$, let $\{\widehat{\psi}_1(x), \dots, \widehat{\psi}_M(x)\}$ be the set of bases generating an approximate feature map $\widehat{\psi} : \Omega \to \widehat{\mathcal{H}}$, and let $\mathcal{X} = \{x_1, \dots, x_N\}$, $x_i \in \Omega$. Let $K \in \mathbb{R}^{N \times M}$ be the observation matrix, where $K_{ij} := \widehat{\psi}_j(x_i)$. For each row $K_{(i)} := \left[\, \widehat{\psi}_1(x_i) \cdots \widehat{\psi}_M(x_i) \,\right]$, define the set $\mathcal{I}_{(i)} := \{\iota_1^{(i)}, \iota_2^{(i)}, \dots, \iota_{M_i}^{(i)}\}$ to be the indices in the observation matrix row $i$ which are nonzero. Then if $\bigcup_{i \in \{1, \dots, N\}} \mathcal{I}^{(i)} = \{1, 2, \dots, M\}$, we denote $K$ as a* shaded observation matrix *(see Figure 2a).*

This definition seems quite abstract, so the following remark considers a more concrete example.

**Remark 1.** *let $\widehat{\psi}$ be generated by the dictionary given by $\mathcal{C} = \{c_1, \dots, c_M\}$, $c_i \in \Omega$. Note that since $\widehat{\psi}_j(x_i) = \langle \psi(x_i), \psi(c_j) \rangle_{\mathcal{H}} = k(x_i, c_j)$, $K$ is the kernel matrix between $\mathcal{X}$ and $\mathcal{C}$. For the kernel*

*matrix to be shaded thus implies that there does not exist an atom $\psi(c_j)$ such that the projections $\langle\psi(x_i), \psi(c_j)\rangle_{\mathcal{H}}$ vanish for all $x_i$, $1 \leq i \leq N$. Intuitively, the shadedness property requires that the sensor locations $x_i$ are privy to information propagating from every $c_j$. As an example, note that, in principle, for the Gaussian kernel, a single row generates a shaded kernel matrix[2].*

Before we prove Proposition 1, recall that a Jordan decomposition of a matrix $\widehat{A} \in \mathbb{R}^{M \times M}$ with no repeated eigenvalues can be composed of $O$ Jordan blocks, where $O$ can be strictly less than $M$. This is in direct contrast to the case where the matrix $\widehat{A}$ has an eigenvalue decomposition, in which case no repeated eigenvalues implies that $O = M$.

**Proposition 1.** *Given $k : \Omega \times \Omega \to \mathbb{R}$ positive-definite on a domain $\Omega$, let $\{\widehat{\psi}_1(x), \ldots, \widehat{\psi}_M(x)\}$ be the set of bases generating an approximate feature map $\widehat{\psi} : \Omega \to \widehat{\mathcal{H}}$, and let $\mathcal{X} = \{x_1, \ldots, x_N\}$, $x_i \in \Omega$. Consider the discrete linear system on $\widehat{\mathcal{H}}$ given by the evolution and measurement equations (3). Suppose that a full-rank Jordan decomposition of $\widehat{A} \in \mathbb{R}^{M \times M}$ of the form $\widehat{A} = P\Lambda P^{-1}$ exists, where $\Lambda = \begin{bmatrix} \Lambda_1 & \cdots & \Lambda_O \end{bmatrix}$, and there are no repeated eigenvalues. Then, given a set of time instances $\Upsilon = \{\tau_1, \tau_2, \ldots, \tau_L\}$, and a set of sampling locations $\mathcal{X} = \{x_1, \ldots, x_N\}$, the system (3) is observable if the observation matrix $K_{ij}$ is shaded according to Definition 1, $\Upsilon$ has distinct values, and $|\Upsilon| \geq M$.*

*Proof.* To begin, consider a system where $\widehat{A} = \Lambda$, with Jordan blocks $\{\Lambda_1, \Lambda_2, \ldots, \Lambda_O\}$ along the diagonal. Then $\widehat{A}^{\tau_i} = \text{diag}(\begin{bmatrix} \Lambda_1^{\tau_i} & \Lambda_2^{\tau_i} & \cdots & \Lambda_O^{\tau_i} \end{bmatrix})$. We have that

$$\mathcal{O}_{\Upsilon} = \underbrace{\begin{bmatrix} K\widehat{A}^{\tau_1} \\ \cdots \\ K\widehat{A}^{\tau_L}. \end{bmatrix}}_{\mathcal{O}_{\Upsilon} \in \mathbb{R}^{NL \times M}}$$

We need to prove that the column rank of $\mathcal{O}_{\Upsilon}$ is $M$, which is not immediately obvious since typically $N \ll M$. To prove the statement, we will show that computing the rank of $\mathcal{O}_{\Upsilon}$ is equivalent to the rank computation of the product of two simple matrices. In what follows, we use the notation $\mathbf{0}_{\mathbb{R}^{I \times J}}$ to denote an $I \times J$ matrix of all zeros.

In the first step, we write the above matrix as the product of two matrices. Then it can be shown that $\mathcal{O}_{\Upsilon}$ is the product of two block matrices

$$\mathcal{O}_{\Upsilon} = \underbrace{\begin{bmatrix} K & \mathbf{0}_{\mathbb{R}^{N \times M}} & \cdots & \mathbf{0}_{\mathbb{R}^{N \times M}} \\ \mathbf{0}_{\mathbb{R}^{N \times M}} & K & \cdots & \mathbf{0}_{\mathbb{R}^{N \times M}} \\ \vdots & \vdots & \ddots & \vdots \\ \mathbf{0}_{\mathbb{R}^{N \times M}} & \mathbf{0}_{\mathbb{R}^{N \times M}} & \cdots & K \end{bmatrix}}_{\widehat{\mathbf{K}} \in \mathbb{R}^{NL \times ML}} \underbrace{\begin{bmatrix} \Lambda_1^{\tau_1} & 0 & \cdots & 0 \\ 0 & \Lambda_2^{\tau_1} & \cdots & 0 \\ \vdots & \vdots & \ddots & \vdots \\ 0 & 0 & \cdots & \Lambda_O^{\tau_1} \\ \hline \vdots & \vdots & \ddots & \vdots \\ \Lambda_1^{\tau_L} & 0 & \cdots & 0 \\ 0 & \Lambda_2^{\tau_L} & \cdots & 0 \\ \vdots & \vdots & \ddots & \vdots \\ 0 & 0 & \cdots & \Lambda_O^{\tau_L} \end{bmatrix}}_{\widehat{\mathbf{A}} \in \mathbb{R}^{ML \times M}}.$$

We need to simplify $\widehat{\mathbf{K}}$ even further. Recall that a matrix's rank is preserved under a product with an invertible matrix. Design a matrix of elementary row operations $U \in \mathbb{R}^{N \times N}$ such that $\check{K} := UK$ is a matrix with at least one row vector of nonzeros; this can be achieved by having an elementary matrix that adds rows together. By the shadedness assumption, such a matrix exists. We can write this operation as

$$UK = \begin{bmatrix} \check{K}_{11} & \check{K}_{12} & \cdots & \check{K}_{1M} \\ \check{K}_{21} & \check{K}_{22} & \cdots & \check{K}_{2M} \\ \vdots & \vdots & \ddots & \vdots \\ \check{K}_{N1} & \check{K}_{N2} & \cdots & \check{K}_{NM}. \end{bmatrix}$$

Without loss of generality, and abusing notation slightly, let this multiplication lead to one nonzero row, with the rest of the elements of the matrix being zero, as

$$
UK = \begin{bmatrix} k_{11} & k_{12} & \cdots & k_{1M} \\ 0 & 0 & \cdots & 0 \\ \vdots & \vdots & \ddots & \vdots \\ 0 & 0 & \cdots & 0 \end{bmatrix}.
$$

Since elementary matrices are full-rank, we then have that $\mathrm{rank}(UK) = \mathrm{rank}(K)$.

To analyze the rank of $\mathcal{O}_\Upsilon$, we apply these elementary matrices to every $K \in \widehat{\mathbf{K}}$. To do so, consider the block-diagonal matrix $\mathcal{U} \in \mathbb{R}^{NL \times NL}$ with $U \in \mathbb{R}^{N \times N}$ along the diagonal, and zeros everywhere else. It can be shown that $\mathcal{U}$ is full-rank, i.e. has rank $NL$. Going back to the observability matrix, we have that

$$
\mathcal{U}\mathcal{O}_\Upsilon = \underbrace{\begin{bmatrix} U & \mathbf{0}_{\mathbb{R}^{N \times N}} & \cdots & \mathbf{0}_{\mathbb{R}^{N \times N}} \\ \mathbf{0}_{\mathbb{R}^{N \times N}} & U & \cdots & \mathbf{0}_{\mathbb{R}^{N \times N}} \\ \vdots & \vdots & \ddots & \vdots \\ \mathbf{0}_{\mathbb{R}^{N \times N}} & \mathbf{0}_{\mathbb{R}^{N \times N}} & \cdots & U \end{bmatrix}}_{\mathcal{U} \in \mathbb{R}^{NL \times NL}} \underbrace{\begin{bmatrix} K & \mathbf{0}_{\mathbb{R}^{N \times M}} & \cdots & \mathbf{0}_{\mathbb{R}^{N \times M}} \\ \mathbf{0}_{\mathbb{R}^{N \times M}} & K & \cdots & \mathbf{0}_{\mathbb{R}^{N \times M}} \\ \vdots & \vdots & \ddots & \vdots \\ \mathbf{0}_{\mathbb{R}^{N \times M}} & \mathbf{0}_{\mathbb{R}^{N \times M}} & \cdots & K \end{bmatrix}}_{\widehat{\mathbf{K}} \in \mathbb{R}^{NL \times ML}} \underbrace{\begin{bmatrix} \Lambda_1^{\tau_1} & 0 & \cdots & 0 \\ 0 & \Lambda_2^{\tau_1} & \cdots & 0 \\ \vdots & \vdots & \ddots & \vdots \\ 0 & 0 & \cdots & \Lambda_O^{\tau_1} \\ \hline \vdots & \vdots & \ddots & \vdots \\ \Lambda_1^{\tau_L} & 0 & \cdots & 0 \\ 0 & \Lambda_2^{\tau_L} & \cdots & 0 \\ \vdots & \vdots & \ddots & \vdots \\ 0 & 0 & \cdots & \Lambda_O^{\tau_L} \end{bmatrix}}_{\widehat{\mathbf{A}} \in \mathbb{R}^{ML \times M}}
$$

$$
= \underbrace{\begin{bmatrix} UK & \mathbf{0}_{\mathbb{R}^{N \times M}} & \cdots & \mathbf{0}_{\mathbb{R}^{N \times M}} \\ \mathbf{0}_{\mathbb{R}^{N \times M}} & UK & \cdots & \mathbf{0}_{\mathbb{R}^{N \times M}} \\ \vdots & \vdots & \ddots & \vdots \\ \mathbf{0}_{\mathbb{R}^{N \times M}} & \mathbf{0}_{\mathbb{R}^{N \times M}} & \cdots & UK \end{bmatrix}}_{\mathcal{U}\widehat{\mathbf{K}} \in \mathbb{R}^{NL \times ML}} \underbrace{\begin{bmatrix} \Lambda_1^{\tau_1} & 0 & \cdots & 0 \\ 0 & \Lambda_2^{\tau_1} & \cdots & 0 \\ \vdots & \vdots & \ddots & \vdots \\ 0 & 0 & \cdots & \Lambda_O^{\tau_1} \\ \hline \vdots & \vdots & \ddots & \vdots \\ \Lambda_1^{\tau_L} & 0 & \cdots & 0 \\ 0 & \Lambda_2^{\tau_L} & \cdots & 0 \\ \vdots & \vdots & \ddots & \vdots \\ 0 & 0 & \cdots & \Lambda_O^{\tau_L} \end{bmatrix}}_{\widehat{\mathbf{A}} \in \mathbb{R}^{ML \times M}},
$$

since $\mathbf{0}_{\mathbb{R}^{N \times N}} \mathbf{0}_{\mathbb{R}^{N \times M}} = \mathbf{0}_{\mathbb{R}^{N \times M}}$. Due to the fact that $\mathrm{rank}(\mathcal{U}\mathcal{O}_\Upsilon) = \mathrm{rank}(\mathcal{O}_\Upsilon)$, we can therefore perform our rank analysis on the simpler matrix $\mathrm{rank}(\mathcal{U}\mathcal{O}_\Upsilon)$. Note that

$$
UK\widehat{A}^{\tau_j} = \begin{bmatrix} k_{11} & k_{12} & \cdots & k_{1M} \\ 0 & 0 & \cdots & 0 \\ \vdots & \vdots & \ddots & \vdots \\ 0 & 0 & \cdots & 0 \end{bmatrix} \widehat{A}^{\tau_j}
$$

$$
= \begin{bmatrix} k_{11}\lambda_1^{\tau_j} & \binom{\tau_j}{1}\lambda_1^{\tau_j - 1} + k_{12}\lambda_1^{\tau_j} & \cdots & k_{1M}\lambda_O^{\tau_j} \\ 0 & 0 & \cdots & 0 \\ \vdots & \vdots & \ddots & 0 \\ 0 & 0 & \cdots & 0 \end{bmatrix}.
$$

Therefore, following some more elementary row operations encoded by $V \in \mathbb{R}^{ML \times ML}$, we have that

$$
V\mathcal{U}\mathcal{O}_\Upsilon = \begin{bmatrix}
k_{11}\lambda_1^{\tau_1} & \cdots & k_{1M}\lambda_O^{\tau_1} \\
k_{11}\lambda_1^{\tau_2} & \cdots & k_{1M}\lambda_O^{\tau_2} \\
\vdots & \ddots & 0 \\
k_{11}\lambda_1^{\tau_L} & \cdots & k_{1M}\lambda_O^{\tau_L} \\
\mathbf{0}_{\mathbb{R}^{M(L-1) \times 1}} & \cdots & \mathbf{0}_{\mathbb{R}^{M(L-1) \times 1}}
\end{bmatrix}
$$
$$
= \begin{bmatrix}
\mathbf{\Phi} \\
\mathbf{0}_{\mathbb{R}^{M(L-1) \times M}}
\end{bmatrix}.
$$

If the individual entries $k_{1i}$ are nonzero, and the Jordan block diagonals have nonzero eigenvalues, the columns of $\mathbf{\Phi}$ become linearly independent. Therefore, if $L \geq M$, the column rank of $\mathcal{O}_\Upsilon$ is $M$, which results in an observable system.

To extend this proof to matrices $\widehat{A} = P\Lambda P^{-1}$, note that

$$
\mathcal{O}_\Upsilon = \begin{bmatrix}
K\widehat{A}^{\tau_1} \\
\cdots \\
K\widehat{A}^{\tau_L}
\end{bmatrix}
$$
$$
= \begin{bmatrix}
KP\Lambda^{\tau_1}P^{-1} \\
\cdots \\
KP\Lambda^{\tau_L}P^{-1}.
\end{bmatrix}
$$
$$
= \widehat{\boldsymbol{K}}\boldsymbol{P}\boldsymbol{\Lambda}^t\boldsymbol{P}^{-1},
$$

where $\boldsymbol{P} \in \mathbb{R}^{ML \times ML}$, $\boldsymbol{\Lambda}^t \in \mathbb{R}^{ML \times ML}$, and $\boldsymbol{P}^{-1} \in \mathbb{R}^{ML \times ML}$ are the block diagonal matrices associated with the system. Since $\boldsymbol{P}$ is an invertible matrix, the conclusions about the column rank drawn before still hold, and the system is observable. $\square$

When the eigenvalues of the system matrix are repeated, it is not enough for $K$ to be shaded. The next proposition proves a lower bound on the number of sampling locations required.

**Proposition 2.** *Suppose that the conditions in Proposition 1 hold, with the relaxation that the Jordan blocks $\begin{bmatrix} \Lambda_1 & \cdots & \Lambda_O \end{bmatrix}$ may have repeated eigenvalues (i.e. $\exists \Lambda_i$ and $\Lambda_j$ s.t. $\lambda_i = \lambda_j$). Then there exist kernels $k(x, y)$ such that the lower bound $\ell$ on the number of sampling locations $N$ is given by the cyclic index of $\widehat{A}$.*

*Proof.* We first prove the lower bound. Pick the Gaussian kernel in the dictionary of atoms framework, with sampling locations $x_i \in \mathcal{X}$ and centers $c_j \in \mathcal{C}$, with the additional property that $x_i \neq x_j \forall i\{1, \ldots, N\}$, $j\{1, \ldots, M\}$. In this case, $\mathbf{K}$ has $\ell - 1$ nonzero, linearly independent rows, and can be written as

$$
\mathbf{K} = \begin{bmatrix}
k_{11} & k_{12} & \cdots & k_{1M} \\
\vdots & \vdots & \cdots & \vdots \\
k_{(\ell-1)1} & k_{(\ell-1)2} & \cdots & k_{(\ell-1)M}
\end{bmatrix}.
$$

Since the cyclic index is $\ell$, this implies that at least one eigenvalue, say $\lambda$, has $\ell$ Jordan blocks. Define indices $j_1, j_2, \ldots, j_\ell \in \{1, 2, \ldots, M\}$ as the columns corresponding to the leading entries of the $\ell$ Jordan blocks corresponding to $\lambda$. WLOG, let $j_1 = 1$. Using ideas similar to the last proof, we can write the observability matrix as

$$
\mathcal{O}_\Upsilon := \begin{bmatrix}
k_{11}\lambda^{\tau_1} & \cdots & k_{1j_\ell}\lambda^{\tau_1} & \cdots \\
\vdots & \ddots & \vdots & \ddots \\
k_{11}\lambda^{\tau_L} & k_{1j_\ell}\lambda^{\tau_L} & \cdots & \\
\vdots & \ddots & \vdots & \ddots \\
k_{(\ell-1)1}\lambda^{\tau_1} \cdots & k_{(\ell-1)j_\ell}\lambda^{\tau_1} & \cdots & \\
\vdots & \ddots & \vdots & \ddots \\
k_{(\ell-1)1}\lambda^{\tau_L} & \cdots & k_{(\ell-1)j_\ell}\lambda^{\tau_L} & \cdots
\end{bmatrix}.
$$

Define $\boldsymbol{\lambda} := [\lambda^{\tau_1} \quad \lambda^{\tau_2} \quad \cdots \lambda^{\tau_L}]^T$. Then the above matrix becomes

$$\mathcal{O}_{\Upsilon} := \begin{bmatrix} k_{11}\boldsymbol{\lambda} & \cdots & k_{1j_2}\boldsymbol{\lambda} & \cdots & k_{1j_\ell}\boldsymbol{\lambda} & \cdots \\ \vdots & \ddots & \vdots & \ddots & \vdots & \ddots \\ k_{(\ell-1)1}\boldsymbol{\lambda} & \cdots & k_{(\ell-1)j_2}\boldsymbol{\lambda} & \cdots & k_{(\ell-1)j_\ell}\boldsymbol{\lambda} & \cdots \end{bmatrix}.$$

We need to show that one of the columns above can be written in terms of the others. This is equivalent to solving the linear system

$$\begin{bmatrix} k_{1j_1} \\ k_{2j_1} \\ \vdots \\ k_{(\ell-1)j_1} \end{bmatrix} = \begin{bmatrix} k_{1j_2} & \cdots & k_{1j_\ell} \\ k_{2j_2} & \cdots & k_{2j_\ell} \\ \vdots & \ddots & \vdots \\ k_{(\ell-1)j_2} & \cdots & k_{(\ell-1)j_\ell} \end{bmatrix} \begin{bmatrix} c_1 \\ c_2 \\ \vdots \\ c_{(\ell-1)} \end{bmatrix}.$$

Since the kernel matrix on the RHS is generated from the Gaussian kernel, from [1], it's known that every principal minor of a Gaussian kernel matrix is invertible, which implies that $\mathcal{O}_{\Upsilon}$ cannot be observable. $\square$

Section 2 gives a concrete example to build intuition regarding this lower bound. We now show how to construct a matrix $\widetilde{K}$ corresponding to the lower bound $\ell$.

**Proposition 3.** *Given the conditions stated in Proposition 2, it is possible to construct a measurement map $\widetilde{K} \in \mathbb{R}^{\ell \times M}$ for the system given by (3), such that the pair $(\widetilde{K}, \widehat{A})$ is observable.*

*Proof.* The construction of the measurement map $\widetilde{K}$ is based on the rational canonical structure of $\widehat{A}^T$ (discussed in section 1), which decomposes $\mathcal{V}$ into $\widehat{A}^T$-cyclic direct summands such that $\mathcal{V} = \mathcal{V}_1 \oplus \cdots \oplus \mathcal{V}_\ell$, where $\ell$ is the cyclic index of $\widehat{A}$ as defined in Proposition 2. Let $\xi_v$ be the minimal polynomial (m.p.) of $v$ (relative to $\widehat{A}^T$): it is then the unique monic polynomial of least degree such that $\xi_v(\widehat{A}^T)v = 0$. Let $\alpha_1(\lambda)$ be the m.p. of $\widehat{A}^T_{|\mathcal{V}_1}$: then $\deg(\alpha_1(\lambda)) < M$. By the rational canonical structure theorem [4], there exists a vector $\widehat{v}_1$, such that $\xi_{v_1}(\lambda) = \alpha_1(\lambda)$. Similarly there exists a vector $\widehat{v}_2$, such that $\xi_{v_2}(\lambda) = \alpha_2(\lambda)$, where $\alpha_2(\lambda)$, is the minimal polynomial of $\widehat{A}^T_{|\mathcal{V}_2}$ and so on. Thus we can obtain $\ell$ such vectors that form the measurement map $\widetilde{K} = [\widehat{v}_1, \widehat{v}_2, \cdots, \widehat{v}_\ell]^T$. Construction of these vectors $\widehat{v}_i$, can be simplified by first performing the Jordan decomposition as $\widehat{A}^T = P\Lambda P^{-1}$. Then the vectors $\widetilde{v}_i$, $i \in \ell$ for $\Lambda$, can be constructed such that the entries corresponding to the leading entries of Jordan blocks of $\Lambda_{|\mathcal{V}_i}$ are nonzero. Such a construction ensures that the m.p. of vector $\widetilde{v}_i$ w.r.t $\Lambda_{|\mathcal{V}_i}$, is also the corresponding m.p. of $\Lambda_{|\mathcal{V}_i}$. Hence the required map is given by $\widetilde{K} = [\widetilde{v}_1, \widetilde{v}_2, \ldots, \widetilde{v}_\ell]^T P^{-1}$. $\square$

The construction provided in the proof of Proposition 3 is utilized in Algorithm 1, which uses the rational canonical structure of $\widehat{A}$ to generate a series of vectors $v_i \in \mathbb{R}^M$, whose iterations $\{v_1, \ldots, \widehat{A}^{m_1-1}v_1, \ldots, v_\ell, \ldots, \widehat{A}^{m_\ell-1}v_\ell\}$ generate a basis for $\mathbb{R}^M$ (see Section 1). Unfortunately, the measurement map $\widetilde{K}$, being an abstract construction unrelated to the kernel, does not directly select $\mathcal{X}$. We will show how to use the measurement map to guide a search for $\mathcal{X}$ in Remark 2. For now, we state a sufficient condition for observability of a general system.

**Theorem 1.** *Suppose that the conditions in Proposition 1 hold, with the relaxation that the Jordan blocks $[\Lambda_1 \quad \Lambda_2 \quad \cdots \quad \Lambda_O]$ may have repeated eigenvalues. Let $\ell$ be the cyclic index of $\widehat{A}$. We define*

$$\mathbf{K} = \begin{bmatrix} K^{(1)T} & \ldots & K^{(\ell)T} \end{bmatrix}^T \tag{1}$$

*as the $\ell$-shaded matrix which consists of $\ell$ shaded matrices with the property that any subset of $\ell$ columns in the matrix are linearly independent from each other. Then system (3) is observable if $\Upsilon$ has distinct values, and $|\Upsilon| \geq M$.*

*Proof.* A cyclic index of $\ell$ for this system implies that there exists an eigenvalue $\lambda$ that's repeated $\ell$ times. We prove the theorem for repeated eigenvalues of dimension 1: the same statement can

---
**Algorithm 1** Measurement Map $\widetilde{K}$

---

**Input:** $\widehat{A} \in \mathbb{R}^{M \times M}$
Compute Frobenius canonical form, such that $C = Q^{-1}\widehat{A}^T Q$. Set $C_0 := C$, and $M_0 := M$.
**for** $i = 1$ **to** $\ell$ **do**
    Obtain MP $\alpha_i(\lambda)$ of $C_{i-1}$. This returns associated indices $\mathcal{J}^{(i)} \subset \{1, 2, \dots, M_{i-1}\}$.
    Construct vector $v_i \in \mathbb{R}^M$ such that $\xi_{v_i}(\lambda) = \alpha_i(\lambda)$ .
    Use indices $\{1, 2, \dots, M_{i-1}\} \setminus \mathcal{J}^{(i)}$ to select matrix $C_i$. Set $M_i := |\{1, 2, \dots, M_{i-1}\} \setminus \mathcal{J}^{(i)}|$
**end for**
Compute $\mathring{K} = [v_1^T, v_2^T, \dots, v_\ell^T]^T$
**Output:** $\widetilde{K} = \mathring{K}Q^{-1}$

---

be proven for repeated eigenvalues for Jordan blocks using the ideas in the proof of Proposition 1. WLOG, let $\mathbf{K}$ have $\ell$ fully shaded, linearly independent rows, and, assume that the column indices corresponding to this eigenvalue are $\{1, 2, \dots, \ell\}$. Define $\boldsymbol{\lambda}_i := [\lambda_i^{\tau_1} \quad \lambda_i^{\tau_2} \quad \cdots \lambda_i^{\tau_L}]^T$. Then

$$\mathcal{O}_\Upsilon := \begin{bmatrix} k_{11}\boldsymbol{\lambda}_1 & k_{12}\boldsymbol{\lambda}_2 & \cdots & k_{1M}\boldsymbol{\lambda}_M \\ \vdots & \vdots & \ddots & \vdots \\ k_{\ell 1}\boldsymbol{\lambda}_1 & k_{\ell 2}\boldsymbol{\lambda}_2 & \cdots & k_{\ell M}\boldsymbol{\lambda}_M \end{bmatrix}.$$

Let $\boldsymbol{\lambda}_1 = \boldsymbol{\lambda}_2 = \cdots \boldsymbol{\lambda}_\ell := \boldsymbol{\lambda}$. Focusing on these first $\ell$ columns of this matrix, this implies that we need to find constants $c_1, c_2, \dots, c_{\ell-1}$ such that

$$\begin{bmatrix} k_{11} \\ \vdots \\ k_{\ell 1} \end{bmatrix} = c_1 \begin{bmatrix} k_{12} \\ \vdots \\ k_{\ell 2} \end{bmatrix} + \cdots + c_{\ell-1} \begin{bmatrix} k_{1\ell} \\ \vdots \\ k_{\ell\ell} \end{bmatrix}.$$

However, these columns are linearly independent by assumption, and thus no such constants exist, implying that $\mathcal{O}_\Upsilon$ is observable. $\square$

While Theorem 1 is a quite general result, the condition that any $\ell$ columns of $\mathbf{K}$ be linearly independent is a very stringent condition. One scenario where this condition can be met with minimal measurements is in the case when the feature map $\widehat{\psi}(x)$ is generated by a dictionary of atoms with the Gaussian RBF kernel evaluated at sampling locations $\{x_1, \dots, x_N\}$ according to (2), where $x_i \in \Omega \subset \mathbb{R}^d$, and $x_i$ are sampled from a non-degenerate probability distribution on $\Omega$ such as the uniform distribution. For a semi-deterministic approach, when the dynamics matrix $\widehat{A}$ is block-diagonal, we can utilize a simple heuristic:

**Remark 2.** *Let $\Omega$ be compact, $\mathcal{C} = \{c_1, \dots, c_M\}$, $c_i \in \Omega$, and let the approximate feature map be defined by (2). Consider the system (3) with $\widehat{A} = \Lambda$, and let $\Upsilon = \{0, 1, \dots, M-1\}$. Then the measurement map $\widetilde{K}$'s values lie in $\{0, 1\}$; in particular, each row $\widetilde{K}^{(j)}$, $j \in \{1, \dots, \ell\}$, corresponds to a subspace $\widehat{\mathcal{H}}_j$, generated by a subset of centers $\mathcal{C}^{(j)} \subset \mathcal{C}$. Generate samples $x_i^{(j)}$ to create a kernel matrix $K^{(j)}$ that is shaded only with respect to centers $\mathcal{C}^{(j)}$. Once this is done, move on to the next subspace $\widehat{\mathcal{H}}_{j+1}$. When all $\ell$ rows of $\widetilde{K}$ are accounted for, construct the matrix $\mathbf{K}$ as in (1). Then the resulting system $(\mathbf{K}, \widehat{A})$ is observable.*

This heuristic is formalized in Algorithm 2. Note that in practice, the matrix $\widehat{A}$ needs to be inferred from measurements of the process $f_\tau$. If no assumptions are placed on $\widehat{A}$, it's clear that at least $M$ sensors are required for the system identification phase. Future work will study the precise conditions under which system identification is possible with less than $M$ sensors.

## 5   Numerical Computation of the Canonical Forms

Realizing the minimum number of sensors relies on the rational canonical form, which can be computed using the Jordan canonical form. Computing the Jordan normal form can be extremely expensive: older algorithms report computation times of the order of $\mathcal{O}(M^5 \mathbb{M}(M))$ [2], where $(\mathbb{M}(M)$

---

**Algorithm 2** Sampling locations set $\mathcal{X}$

---

**Input:** $\widehat{A} = C$, lower bound $\ell$
Decompose $C$ to generate invariant subspaces $\widehat{\mathcal{H}}_j$, $j \in \{1, 2, \ldots, \ell\}$ (see section 1)
**for** $j = 1$ **to** $\ell$ **do**
    Obtain centers $\mathcal{C}^{(j)}$ w.r.t subspace $\widehat{\mathcal{H}}_j$,
    Generate samples $x_i^{(j)}$ to create a kernel matrix $K^{(j)}$ that is shaded only with respect to centers $\mathcal{C}^{(j)}$
**end for**
**Output:** Sampling locations set $\mathcal{X} = \{x^{(1)}, x^{(2)} \cdots, x^{(l)}\}$.

---

---

**Algorithm 3** Kernel Observer (Transition Learning)

---

**Input:** Kernel $k$, basis centers $\mathcal{C}$, final time step $T$.
**while** $\tau \leq T$ **do**
    1) Sample data $\{y_\tau^i\}_{i=1}^M$ from $f_\tau$.
    2) Estimate $\widehat{w}_\tau$ via standard kernel inference procedure.
    3) Store weights $\widehat{w}_\tau$ in matrix $\mathcal{W} \in \mathbb{R}^{M \times T}$.
**end while**
To infer $\widehat{A}$, define matrix $\Phi = \mathcal{W}^T \mathcal{W}$. Then:
**for** $i = 1$ **to** $M$ **do**
    At step $i$, solve system

$$\widehat{A}^{(i)} = \left( (\Phi + \lambda I)^{-1} \left( \mathcal{W}^T \mathcal{W}^{(i)} \right) \right)^T, \tag{2}$$

    where $\widehat{A}^{(i)}$, and $\mathcal{W}^{(i)}$ are the $i$th columns of $\widehat{A}$ and $\mathcal{W}^{(i)}$ respectively.
**end for**
Compute the covariance matrix $\widehat{B}$ of the observed weights $\mathcal{W}$.
**Output:** estimated transition matrix $\widehat{A}$, predictive covariance matrix $\widehat{B}$.

---

**Algorithm 4** Kernel Observer (Monitoring and Prediction)

---

**Input:** Kernel $k$, basis centers $\mathcal{C}$, estimated system matrix $\widehat{A}$, estimated covariance matrix $\widehat{B}$.
**Compute Observation Matrix:** Compute the cyclic index $\ell$ of $\widehat{A}$, and compute $K$.
**Initialize Observer:** Use $\widehat{A}$, $\widehat{B}$, and $K$ to initialize a state-observer (e.g. Kalman filter (KF)) on $\widehat{\mathcal{H}}$.
**while** measurements available **do**
    1) Sample data $\{y_\tau^i\}_{i=1}^N$ from $f_\tau$.
    2) Propagate KF estimate $\widehat{w}_\tau$ forward to time $\tau + 1$, correct using measurement feedback with $\{y_{\tau+1}^i\}_{i=1}^N$.
    3) Output predicted function $\widehat{f}_{\tau+1}$ of KF.
**end while**

---

denotes operations sufficient to multiply two $M \times M$ matrices over $\mathbb{R}$): iterative approximation algorithms such as the one in [3] can be used to reduce the computation time. Algorithms also exist for computing approximate Jordan decompositions. We have examined strategies for avoiding this computation by directly considering the invariant subspaces obtained from eigendecompositions of the dynamics matrix $\widehat{A}$: this algorithm will be reported in future work.

<div align="center">Table 2: Notations</div>

| Notations | |
|---|---|
| $\mathcal{A}$ | Linear transition operator in the RKHS $\mathcal{H}$ |
| $\widehat{A}$ | Linear transition operator in the strict subspace $\widehat{\mathcal{H}}$ of RKHS $\mathcal{H}$ |
| $\alpha(\lambda)$ | Minimal polynomial |
| $\alpha_i(\lambda)$ | Minimal polynomial w.r.t $\Lambda_{i-1}$ or $\mathcal{A}_{|\mathcal{V}_i}$ |
| $\mathcal{B}$ | Banach space |
| $\mathcal{C}$ | Array of basis centers generating a finite-dimensional covering of $\mathcal{H}$ |
| $\mathcal{C}^{(j)}$ | Subset of basis centers $\mathcal{C}$ |
| $c_i$ | Basis center, $i^{th}$ element of $\mathcal{C}$ |
| $f_\tau$ | Mean of time-varying stochastic process at instant $\tau$ |
| $\mathcal{H}$ | Reproducing Kernel Hilbert Space (RKHS) |
| $\widehat{\mathcal{H}}$ | Approximate Reproducing Kernel Hilbert Space |
| $\widehat{\mathcal{H}}_j$ | $j^{th}$ subspace of $\widehat{\mathcal{H}}$ |
| $k(\cdot,\cdot)$ | Positive-definite kernel on a domain $\Omega$ |
| $\widetilde{K}$ | Measurement map $\in \mathbb{R}^{\ell \times M}$ |
| $\mathring{K}$ | Measurement map $\in \mathbb{R}^{\ell \times M}$ corresponding to Jordan normal form $\Lambda$ |
| $\mathcal{K}$ | Linear measurement operator that maps $\mathcal{H} \to \mathbb{R}^N$ |
| $K$ | Kernel matrix between the data points and basis vectors |
| $\ell$ | Lower bound on sampling locations, the cyclic index of a matrix |
| $L$ | Total number of time instances, at which measurement or samples are taken. |
| $\mathcal{L}$ | Linear operator in Banach space $\mathcal{B}$ |
| $M$ | Number of atoms in $\widehat{\mathcal{H}}$ |
| $N$ | Number of sensing or sampling locations |
| $\Lambda$ | Jordan normal form |
| $\mathcal{O}_\Upsilon$ | Observability Matrix |
| $\Upsilon$ | Set of instances $\tau_i$ |
| $P$ | Similarity transformation matrix |
| $\psi(\cdot)$ | Smooth map $\psi : \Omega \to \mathcal{H}$ |
| $\tau$ | Discrete time index |
| $u$ | a function in Banach space $\mathcal{B}$ |
| $\mathcal{V}$ | Linear space |
| $w_\tau$ | Weight vector $\in \mathbb{R}^M$ at instant $\tau$ |
| $w_0$ | Initial weight vector |
| $\widehat{w}_\tau$ | Estimate of $w_\tau$ |
| $x_i$ | $i^{th}$ sensing or sampling location |
| $x_i^{(j)}$ | $i^{th}$ sensing or sampling location w.r.t $\mathcal{C}^{(j)} \subset \mathcal{C}$ |
| $x^{(j)}$ | Sensing or sampling locations w.r.t $\mathcal{C}^{(j)} \subset \mathcal{C}$ |
| $\mathcal{X}$ | Set of sampling or sensing locations |

## Footnotes

[1]In general, the subspaces $\mathcal{V}_i$ are not unique for a fixed $\mathcal{A}$.

[2] However, in this case, the matrix can have many entries that are extremely close to zero, and will probably be very ill-conditioned.