[Reviews · NeurIPS 2016]

Reviewer 1

Summary

This paper introduced a method based on system's theory to tackle the problems of monitoring and predicting the behaviour of complex spatiotemporally evolving phenomena. The proposed methodology was especially indicated to achieve an accurate estimate of the latent forces driving the system's spatiotemporal dynamics by sampling on a reduced number of locations. Good results on illustrative experiments show the validity of the approach.

Qualitative Assessment

This paper introduces an interesting kernel approach to deal with spatiotemporal stochastic dynamical systems. The contribution is well motivated and experimentally validated. There are, however, some concerns that I hope the authors may address: * The field of nonlinear dynamical system analysis with kernels is huge, and some relevant references are missing; authors could check papers by L. Ralaivola and F. d’Alche Buc (kernel kalman filtering), Tuia et al (explicit recursion in Hilbert spaces), Rojo-Alvarez et al. (kernel filtering), M. Mouattamid and R. Schaback (recursive kernels), or M. O. Franz and B. B. Schölkopf (online Volterra approximations). Authors could also consider cite the work on latent force models by M. Alvarez and N. Lawrence in their section 1.1. * Experimental examples are really illustrative and appreciated. - However, I would welcome the authors to include comparison with other sparse approximations recently presented in the GP literature (e.g. KISS-GP, FITC, etc.) mostly based on the (quite related) philosophy of inducing variables and observers, and the adaptive kernel filtering strategies based on budget allocation (Budget-KLMS, KRLS) very common in signal processing. - Few details are given about the experimental use of the different approximation methods revised in 2.1. I understand that only dictionary construction following a minimum reconstruction error of the basis is followed. Could authors confirm and clarify this? How critical is the selection of the basis (inducing points) in tracking dynamics? Is |C| selected a priori or by fixing a max error threshold on the basis reconstruction? - I'd like to receive some more clarification about some issues in the experiment 3.3; (1) figures 6b-c do not show a daily sampling as claimed, it is not clear to me how it is possible that models perform roughly equal at day and night (which is a more challenging); (3) probably related to the previous concern is the fact that 'original GP' using 400000 training points already achieves an average (and quite poor) RMSE of 3K, which makes the KO errors completely useless; and (4) I find kind of adventuresome to conclude that internal dynamics were not captured properly in 2011 because of the poor performance in 2012, since there is at least 4 months in 2012 whenever this is not the case. Some clarifications are needed.

Confidence in this Review

2-Confident (read it all; understood it all reasonably well)


Reviewer 2

Summary

This paper discusses an approach to spatiotemporal modeling. In particular, the authors use kernel methods for formulate the dynamics, and discuss the number of sampling locations required for observability. They derive a lower bound given by the Jordan decomposition of the state transition matrix. Experimental results are shown with symthetic and real-world data sets, showing favorable results for the theoretical observations.

Qualitative Assessment

The paper includes an interesting approach to spatiotemporal modeling and a theoretical contribution on the observation size. I think the discussed approach to the spatotemporal modeling with kernel methods is useful, and the obtained lower bound of the observation size is potentially significant. I feel, however, the focus of the paper is not very clear, as I detail below. - As stated in Sec.2.2, the main contribution of this paper is the lower bound of the samping size necessary for the observability. It is not clearly verified, however, whether or not the theoretical lower bound provides a suitable number in practical problems. In the two examples of Sec. 3.2, the lower bound is two and only the results with two observation points are shown. In Sec. 3.3, the value of lower bound is not shown or used. To verify the usefulness of the theoretical results, experimental results should be shown with different values of N including the theoretical lower bound, and they should be compared. - In Section 2.2 (main results), it is assumed that \hat{A} is known. As stated in the last part of Section 2.2, \hat{A} should be estimated in practice, but this procedure is not at all straightforward. In the second paragraph of Sec. 3.2, a method for obtaining \hat{A} and the system (3) is discussed, and this construction is already very interesting. I recommend the authors should discuss this construction in more detail. The paper could even focus this part rather than the lower bound of the sampling size, from my viewpoint. - Minor comments: In line 63: define the meaning of "nonseparable".

Confidence in this Review

2-Confident (read it all; understood it all reasonably well)


Reviewer 3

Summary

The paper proposes a kernel observer framework for spatiotemporal modeling. The model combines kalman filters to random fourier features and sparse GPs.

Qualitative Assessment

The paper proposes a "kernel observer" framework for spatiotemporal modeling. The paper is somewhat hard to read and tries to perhaps cover too much material. My biggest problem with the paper is that the proposed model is not properly defined, and the model learning is completely ignored in the main text. The kalman filter framework is described in 2.1. and the experimental sections names a few more steps, but it's still unclear how the model works. It seems that the key is how the sparse GP "generates the weights", but that is not defined, nor is what sparse GP is used. Is the GP over time or space? The paper reads as their combination of RFF, kernel kalman filters and sparse GPs are general knowledge that does not require much explanation. The main contribution of the paper seems to be theoretical results on observability, but it's difficult to assess how significant they are. The synthetic example illustrates this result, but I don't understand why an angular dataset was chosen instead of spatiotemporal one. Section 3.2. presents a comparison to two competing methods, which are quite consistently outperformed. However, the sparse GP method ("original") seems to almost always outperform all other methods. Wouldn't the "original" method then be a better choice? It's not described why it's a "baseline" (perhaps it only gives training errors?). The introduction claims that the method can detect optimal sensing locations, but I'm not sure if the experiments reflect this. There should be an experiment comparing e.g. grid/random sensing locations and optimal sensing locations as a function of location number. Overall I believe there might be significant findings underlying the paper, but the paper is not up to NIPS quality in presentation.

Confidence in this Review

1-Less confident (might not have understood significant parts)


Reviewer 4

Summary

This paper describes the problem of needing to model a spatiotemporal process using only a small number of physical locations. The problem is rephrased using systems theory to provide a mechanism for identifying the optimal observation locations. Algorithms and associated proofs are provided to define the strategy and bound the minimum required complexity as a function of the complexity of the evolution of the system. This is compared to nonstationary kernel-based approximation methods in numerical experiments which show significant improvements in training time.

Qualitative Assessment

Overall, not bad. I think many of the paragraphs are far too long, and that often hurts the readability, but I think that the content is solid. The introduction of Banach spaces adds nothing, which actually is part of a trend in this paper to add theory where it is not needed. If the presentation could be tightened up some, with more of a practical slant, this paper would be very good. As it is, the reason I gave a Technical score 2 rather than 3 was primarily for the issues with the proofs I list below, and not for the actual discussion in the main paper. The experiments look fine, although Figure 3 is a bit confusing. I have one significant fear about this paper, and that is why I am placing it here rather than toward the end of my comments: all the comments I have ever read about the Jordan canonical form suggest that it is numerical unstable to compute. In a setting where that factorization of an admittedly inaccurate matrix ("Note that in practice, the matrix hat A needs to be inferred ...") is interacting with a kernel matrix which is notoriously subject to ill-conditioning, how can one be certain that the algorithms in this article will not be subject to instability? Some acknowledgement of this should be included, if only to warn potential practitioners, or some proof as to why this will not be an issue. The lack of discussion of numerical issues was the reason for a rating of 3 instead of 4 on the Potential Impact score. I am not an expert in systems theory, and as such I may be way off base here, but I feel like much of the discussion regarding the proofs is unnecessarily abstract. For example, Definition 1 could simply be "K has no zero valued columns", I think. In turn, Remark 1 could simply be "All positive definite kernels which are always positive produce shaded matrices", which would exclude oscillatory or compactly supported kernels, as I think it must. Also, is a 1974 book really the most up to date reference available for cyclic index of a matrix? If it is, maybe there are more popular tools for conducting these proofs? Likewise, the immediate value of Proposition 1 is lost because by that point, the reader may have forgotten the assumption N < < M and be confused as to why one would need to prove that the product of two full rank matrices is full rank (as I did, initially). Also, an important component of the proposition, that L > = M, is stated almost as a throwaway at the end, rather than as a condition of the setting; it would probably make more sense to start with that as an assumption, along with the assumption that Upsilon contains all (or at least M) unique values. Also, I think there's a typo in the proof of proposition 1, where a bold K hat is written on line 83, but never defined or heard from again. And if A hat is full rank with no repeated eigenvalues, I think the Lambda matrix just has M lambda values in it, because the Jordan decomposition would simply be the spectral decomposition? That would be easier to comprehend than the confusing introduction of the Lambda_O term. I also would prefer cleaner notation for the matrix-matrix products in this proof, where the use of [K ... K] implies something other than what (I think) is meant. O_Upsilon is defined as a matrix with NL rows and M columns, but the product written directly beneath it implies that the matrix should have N rows and M columns. That of course can not be true since an NxM matrix with N < < M will not have linearly independent columns. What I think is meant is a block diagonal matrix of size NL x ML with L copies of K along the diagonal. Then the decomposition would be [K [Lambda^tau_1 O_Upsilon = ... ... K] Lambda^tau_L] Or maybe I do not understand what O_Upsilon is, in which case the notation defining it is confusing. And I think the same issue extends to line 88 where the logic is applied to the Jordan decomposition. Also, the Lambda^t notation is not obvious: that matrix is not being raised to the t power? And, actually, the statement about the sizes of P and Lambda^t is not correct or even consistent with the sizes you wrote earlier - here the resulting matrix would be size NxML, which would certainly not have independent columns. The notation for the first part of the proof of proposition 2 is also confusing, but it at least looks consistent. The proof of proposition 3 also seems fine if perhaps a bit confusing - maybe a cleaner presentation in more than 1 paragraph would help? I think it should be possible to show that, with probability 1, the columns of the K matrix, for all positive definite kernels with only positive values, will be linearly independent for points drawn randomly from the domain. This was shown to some degree in the differential equation setting by [Hon et al, On unsymmetric collocation by radial basis functions], and for the approximation setting by [Hangelbroek et al, Nonlinear approximation using Gaussian kernels]. I mention this only to alert the authors - no such proof need appear in this article. Minor points: * Figure 5(b) says that it is a box plot, but does not look like one. * The phrase "much lesser" is used. * K, and variants with hats and twiddles and circles, is used too much. It makes the content hard to discern. * spelling: nonistropic * The last paragraph of page 2 starts with a poorly written sentence

Confidence in this Review

1-Less confident (might not have understood significant parts)


Reviewer 5

Summary

The paper considers the problem of estimating latent state of a spatiotemporally evolving function using few sensory measurements, showing that the usage of a dynamical systems prior over temporal evolution of weights of a kernel model can be implemented without the need for constructing non-stationary kernel functions.

Qualitative Assessment

The paper is technically sound and the experimental results are interesting. However, I am not fully aware of the large body of related works on kernel methods, so I cannot conclusively judge the originality and added value of the contribution.

Confidence in this Review

1-Less confident (might not have understood significant parts)


Reviewer 6

Summary

The paper deals with the interesting problem of spatiotemporal processes modeling and inference. The problem is becoming of great interest in the machine learning community and builds up from classical literature in geostatistics. In particular, the authors are interested in the so called monitoring problem. That is to say, estimate the latent state of a process of interest using as least sensors as possible, combined with a predictive model for the process. The key point the authors want to stress is the use of the least possible amount of sensors to infer and predict the process state. The contributions are mainly two:­ 1- They show that a good modeling does not necessarily rely on the design of complex spatiotemporal kernel function. In particular, spatiotemporal processes can be effectively modeled by using stationary kernels combined with time varying mixing parameters. 2-­ They provide sufficient conditions on the minimum number of sensors required to get an accurate estimate.

Qualitative Assessment

Overall the paper is well written, theoretically sound and well organized. The problem is interesting and timing. Some minor typos have been found: - line 111: “is an RKHS” —> “is a RKHS” - line 161: in the last entry of the vector y the transpose is in the wrong place The major concerns regard: ­- plots and figure: they are definitely too small and almost not understandable. -­ proof of Proposition 1: I believe there is a dimensional problem in writing the - observability matrix between line 80 and 81. The same problem can be found in the - equation between line 84 and 85. The matrix of K’s shouldn’t be diagonal. Moreover I - have the feeling that the elements of the matrix K\hat{A}^{\tau_j} between line 83 and 84 are wrong. -­ proof of Proposition 2, line 98 I believe that “fully shaded” has not been defined. Is it just 1 shaded? Minor concerns are: ­- equation between line 103 and 104, I believe that the elements in the third column are in the wrong position -­ equation between line 104 and 105, it is not completely clear the full structure of the observability matrix. It would help to write it a little bit more explicitly. A part from these issues, the paper is fair and I believe the contribution is sufficient for acceptance.

Confidence in this Review

2-Confident (read it all; understood it all reasonably well)